# The Impact of the Sweetened Beverages Tax on Their Reformulation in Poland—The Analysis of the Composition of Commercially Available Beverages before and after the Introduction of the Tax (2020 vs. 2021)

**DOI:** 10.3390/ijerph192114464

**Published:** 2022-11-04

**Authors:** Regina Ewa Wierzejska

**Affiliations:** Department of Nutrition and Nutritional Value of Food, National Institute of Public Health NIH—National Research Institute, Chocimska St. 24, 00-791 Warsaw, Poland; rwierzejska@pzh.gov.pl

**Keywords:** sweetened beverages, tax, sugars, reformulation, health

## Abstract

The aim of this study was to estimate changes in the composition of carbonated and non-carbonated sugar-sweetened beverages before and after the introduction of the beverage tax in Poland. Based on the labels of 198 drinks, the composition and nutritional values of the drinks were compared. The nonparametric Mann–Whitney test was applied to compare the differences in the sugar and juice content as well as energy value. After the introduction of the tax, the median sugar content in the carbonated beverages decreased from 8.6 g to 6.9 g/100 mL (*p* = 0.004), while in the non-carbonated beverages, it decreased from 5.5 g to 4.8 g/100 mL (*p* < 0.001). In the entire beverage group, there was a significant drop in the proportion of beverages that contained >5 g of sugars/100 mL (44.4% in 2021 vs. 70.2% in 2020). The median juice content in the carbonated beverages increased from 1.0% to 3.3% (*p* = 0.121), but totalled 20.0% for both periods in the non-carbonated beverages. The percentage of beverages with a tax-exempt composition (juice content ≥ 20% and sugar content ≤ 5 g/100 mL) almost tripled. After the introduction of the tax, beneficial changes in the compositions of 62% of the analysed beverages were observed in terms of their sugar and/or juice content.

## 1. Introduction

Sugar-sweetened beverages (SSBs) belong to food products that the literature points out most frequently as having a negative impact on health, while carbonated soft drinks are classified as ultra-processed foods [1,2]. It is currently known that the health effects of high sugar intake extend far beyond the problem of dental caries. As found in a meta-analysis of prospective cohort studies from 2015, people who drink one portion (250 mL) of SSBs daily have a 13% higher risk of developing type 2 diabetes [3]. In accordance with a meta-analysis of such studies from 2020, one beverage portion increases the risk of developing this disease by 19% and obesity by 12% [4]. However, it cannot be forgotten that disease causes are generally complex, and, according to some authors, the role of SSBs in the development of obesity and diabetes has yet to be fully explored [5,6,7,8,9].

Both the World Health Organization (WHO) and scientific societies pay particular attention to the consumption of simple sugars in their nutritional guidelines. Although there are two different scientific terms referring to sugars, namely ‘added sugars’ and ‘free sugars’, these are often applied interchangeably in publications, and it is recommended that the consumption of such sugars should not exceed 10% of dietary energy [10,11,12,13,14,15,16]. SSBs are considered the highest source of added sugars in many countries [12,17,18]. A worldwide assessment of SSB consumption shows that it is the highest among the residents of Trinidad and Tobago (2.5 servings daily on average), while it is lowest in China (0.05 servings daily on average) [19]. It is worrying that, in many countries, children’s diets include large amounts of sweetened beverages [17,20,21,22,23].

In the face of the global growing problem of obesity, the WHO requires countries to undertake multi-sectoral measures to improve accessibility to healthy food, such as the introduction of the sweetened beverage tax [24]. In some countries, including Poland, legal limits are placed on the sale of sweetened beverages in schools. There are also self-regulations regarding advertising food for children and educational campaigns [25,26]. Other countries also implement front-of-package warning labels [5,15,20,27]. To highlight the necessity of changing eating habits, it is worth invoking the reaction of world-class football player Cristiano Ronaldo, who, during the EURO 2020 conference, moved bottles of a well-known carbonated caffeine beverage away to encourage the drinking of water during the EURO 2020 conference [28]. This event helped visualize the problem explicitly for a large group of young people.

France was the first country to pass a law introducing the tax on SSBs in 2012 [29]. In subsequent years, it became more popular, and it is now in place in more than 40 countries. In Europe, the tax was implemented in countries such as the United Kingdom, Norway and Belgium in North America it was implemented in Mexico and some major cities in the USA; in South America in Chile; in Asia in Saudi-Arabia and Thailand; and in Africa in Morocco [30,31]. In January 2021, Poland joined the group of countries where a tax was imposed not only on beverages with added sugars, but also on beverages with sweeteners, regardless of the natural or synthetic origin of these substances. This approach is aimed at persuading consumers to limit the volumes of sweetened beverages they drink and to encourage them to opt for water. The value of the sweetened beverage tax implemented in Poland depends on beverage composition. The tax charge is fixed, and totals PLN 0.5 per litre of the beverage, if the sugar content does not exceed 5 g/100 mL, or if there is at least one sweetener. Furthermore, for each gram of sugar in excess of 5 g/100 mL, the tax increases by PLN 0.05. A lower tax applies to beverages containing a minimum of 20% juice, while beverages which have at least 20% juice in their composition and for the sugar content does not exceed 5 g/100 mL [32], are fully tax-exempt. This is supposed to work as an incentive for manufacturers to change the composition of their beverages [32].

Due to the relatively short period of time during which the tax has been in operation, there is no available data regarding its long-term impact on consumer behaviour. Analyses from the first years testify to its effectiveness, however [17,33,34]. In Mexico, two years after the introduction of the tax, the SSB sales decreased by 17% [35]; in Philadelphia, where the beverage tax is the highest compared to other US cities, the purchase of taxed beverages decreased by 42% in comparison to Baltimore, where the tax was not applied [36]. In addition, one study found that not only did purchases of SSBs drop significantly in Philadelphia, but sales of bottled water increased at the same time. This could be explained by the fact that the tax there also covers low-calorie beverages, thus eliminating the possibility of choosing them as an alternative to high-sugar beverages [37]. In Poland, the current assessment conducted by the Market Research Centre shows that the decline of the sales of carbonated beverages reached 20%, with an increase in prices by an average of 36% [38]. Based on the literature, it is estimated that an increase in the price of SSBs by 10% results in a 10% decrease in their consumption [39,40], and in light of a recently published meta-analysis of 62 studies conducted worldwide, a tax on SSBs causes a 15% decrease in sales of these beverages [41].

One of the methods of reducing or avoiding the tax is changing the composition of beverages in a healthy way, which was observed in other countries [35,42]. In Poland, to the best of the author’s knowledge, this is the first study of its kind. The aim was to assess the composition of carbonated, sugar-sweetened beverages (CSSBs) and non-carbonated, sugar-sweetened beverages (NCSSBs), commercially available in Warsaw, in the period before and after the introduction of the tax, in terms of the sugar content, using sweeteners, the fruit or vegetable juice content, and their energy value.

## 2. Materials and Methods

### 2.1. Study Design and Data Collection

The study was conducted in 2020 (from August to October) and in 2021 (from August to October) in 5 large supermarkets in Warsaw—Carrefour, Lidl, Biedronka, Kaufland, and Topaz. In 2020, data on the parameters of the nutritional value of the whole range of CSSBs and NCSSBs (excluding beverages labelled as ‘zero’ and ‘no added sugar’; milk-based beverages and functional beverages: isotonic drinks and energy drinks) were collected. In 2021, the collected data only related to the same beverages that were available in the market in 2020 (the same brand, name, and flavour). The data collection took place at shops in some districts of Warsaw, such as Praga Północ, Mokotów, Ursynów, and Centrum.

By photographing labels with nutritional value charts (mandatory on products in Poland), information on the sugar content per 100 mL of the beverage and the energy value was collected. From the label ingredients list, data on the type of sugars and sweeteners used and information on the added juice content were collected. The same beverages that were on offer at the said shops were included only once, as were the same beverages of different unit packaging volumes.

The beverages, commercially available both in 2020 and 2021, were analysed in terms of a change in their composition after the introduction of the sugar tax in Poland.

### 2.2. Statistical Analysis

After verifying the normal data distribution (Shapiro–Wilk test) which showed that the data were not normally distributed, the parameters analysed were presented as the median, minimum and maximum. Due to the various number of carbonated and non-carbonated beverages, comparisons in terms of the sugar, juice content and energy value were carried out only for these two categories, and not for the entire group of beverages. To compare the differences between the two groups, the nonparametric Mann–Whitney test was applied.

## 3. Results

At shops included in the study in 2020, there were 292 ranges of CSSBs and NCSSBs. In 2021, 198 of the same ranges of beverages (67.8% of the 2020 range) remained on offer on the market, of which 88 were carbonated beverages (such as lemonades, orangeades, and cola drinks), and 110 were non-carbonated (such as fruit-flavoured drinks, fruit drinks, fruit and vegetable drinks, nectars, tea-fruit drinks, and tea-herbal drinks).

### 3.1. Sugar Content and Use of Sweeteners

In 2020, the sugar content in the entire analysed group of beverages varied significantly from 1.7 g/100 mL to 13.0 g/100 mL. In CSBBs, the sugar content was higher (median 8.6 g/100 mL) than in NCSSBs (median 5.5 g/100 mL) (*p* < 0.001) (Table 1). The sugar content > 5 g/100 mL that leads to a higher tax charge was traced in 70.2% (139/198) of the beverages, including in 85.2% (75/88) CSSBs and 58.2% (64/110) NCSSBs. In total, 66.7% (132/198) of the beverages were sweetened only with sugars, usually sucrose (40.9%; 81/198 of the beverages), then sucrose and/or glucose-fructose syrup, which are alternative sugars (23.7%; 47/198 of the beverages). The beverages sweetened with both sugars and sweeteners accounted for 24.7% (49/198). Among sweeteners, the most frequently used were acesulfame K (11.1%; 22/198 of the beverages) and steviol glycosides (10.1%; 20/198), while the least frequently used was aspartame (2.5%; 5/198 of the beverages).

In 2021, the range of the sugar content in the whole group of beverages did not change and was 1.7 g/100 mL–13.0 g/100 mL. Both in CSSBs and in NCSSBs, a statistically significant difference in the sugar content was observed. In general, the median of the sugar content in CSSBs decreased to 6.9 g/100 mL (*p* = 0.004), which indicates a decline of 19.8% (1.7 g/100 mL), while in NCSSBs, the median reduced to 4.8 g/100 mL (*p* < 0.001), which is a decrease of 12.7% (0.7 g/100 mL) (Table 1). A higher sugar reduction was observed in beverages whose formulation was changed after the introduction of the tax. These beverages in the CSSB group had, on average, 43.0% less sugars (3.7 g sugars/100 mL), while those in the NCSSB group had 40.2% less (3.3 g sugars/100 mL).

After the introduction of the tax, there was a decline in the proportion of beverages that contained over 5 g of sugars/100 mL, both in the CSSB and NCSSB groups (Figure 1). In the entire group of beverages, there were over 16% beverages less sweetened with sugars only (50.0%; 99/198 in 2021 vs. 66.7%; 132/198 in 2020), and there were over 11% beverages more sweetened with both sugars and sweeteners (36.4%; 72/198 vs. 24.7%; 49/198). Sweeteners alone were used to a small extent (3.5%; 7/198 of beverages). After the taxation of beverages, alternative sweetening with sucrose and/or glucose-fructose syrup also became less popular (9.1%; 18/198 of beverages in 2021 vs. 23.7%; 47/198 in 2020). Among sweeteners, the most frequently used were sucralose (22.2%; 44/198 of beverages) and acesulfame K (12.1%; 24/198), while the least frequently is still aspartame (2.0%; 4/198 of beverages).

### 3.2. Juice Content

In 2020, out of the entire group of beverages, 81.3% (161/198) contained fruit or vegetable juices, while their content in such beverages ranged from 0.01% to 63.0%. There was a small increase in beverages with added juices after the introduction of the tax (84.8%; 168/198) with the minimum and maximum juice content being 0.05% and 80.0%, respectively.

In the CSSB group, the median of the juice content in 2020 amounted to 1.0%. Although it increased to only 3.3% (*p* = 0.121) in 2021, the proportion of beverages with at least 20% juice almost doubled (Figure 1). In NCSSBs, before the tax, the median of the juice content was 20.0% and it remained the same after the tax (Table 1), while the percentage of beverages with at least 20% juice increased only slightly (Figure 1).

### 3.3. Energy Value

In 2020, the energy value in the entire group of beverages ranged from 11.0 kcal/100 mL to 65.0 kcal/100 mL. The energy value of CSSBs, due to the higher sugar content, was higher (median 35.0 kcal/100 mL) than that of NCSSBs (median 27.2 kcal/100 mL) (*p* = 0.002). In 2021, the energy value of both beverage groups decreased, and the differences identified were statistically significant. In the case of CSSBs, the median dropped to 29.0 kcal/100 mL (*p* = 0.006), while in the case of NCSSBs it dropped to 20.0 kcal/100 mL (*p* < 0.001) (Table 1).

### 3.4. Reformulation of Beverages in the Context of the Tax Criteria

After the introduction of the tax, 62.6% (124/198) of the beverages, of which 53.4% (47/88) were CSSBs and 70.0% (77/110) were NCSSBs, had a changed composition in terms of sugar content and/or juice content. There was an increase in the proportion of beverages with ≥20% juice subject to a lower tax charge from 46.5% (92/198) of the beverages to 61.1% (121/198). Moreover, there was nearly a triple increase in the percentage of beverages that were fully tax-exempt (beverages containing ≥20% juice and ≤5 g of sugar/100 mL) (Figure 1).

## 4. Discussion

The analysis conducted shows that the tax has led to a beneficial change in the sugar content. The decrease in the sugar content was higher in the CSSB group (on average by 1.7 g/100 mL) than in the NCSSB group (on average by 0.7 g/100 mL). It can be assumed that this stems from the considerably lower sugar content in NCSSBs before the tax, and thus this small decrease was enough to reach a sugar content (median 4.8 g/100 mL) below the threshold that translated to a lower tax charge. What needs to be emphasised here is the considerable difference in the juice content between these groups of beverages, as well as natural sugars introduced in higher amounts to NCSSBs, which arguably results in the lower amount of sugar added by manufactures. The general decrease in the sugar content did not exceed 20%, which is caused by the fact that the composition of more than 37% of beverages did not change after the introduction of the tax. In the case of beverages reformulated by manufacturers, the sugar content fell more considerably (on average by 3.7 g/100 mL–43% in CSSBs and 3.3 g/100 mL–40% in NCSSBs), which would have allowed the manufacturers to utilise the ‘light’ or ‘reduced sugar’ nutritional claim [43]. The range of beverages sweetened with sweeteners increased only to a small extent (by 15.2%), which can be attributed to the fact that the tax also applies to beverages with sugar substitutes.

Changes in terms of the sugar content implemented in Poland until the time of this study were not as high as those in the United Kingdom, where manufacturers substituted sugar with sweeteners to a higher extent. In 2014, i.e., before the tax, the mean sugar content in carbonated soft drinks amounted to 9.1 g/100 mL in the United Kingdom. However, after introduction of the tax in 2018, it was only 4.4 g/100 mL. Additionally, a comparison of 83 beverage ranges commercially available before and after the tax (a study analogous to ours) showed that the mean sugar content per 100 mL of a beverage decreased by 3.8 g, which accounted for 42% of the overall content [42]. In Columbia, where manufactures reformulated their beverages voluntarily to avoid the tax introduction, an assessment of 36 beverages commercially available in 2016 and 2018 showed that the median of the sugar content fell significantly (from 9.2 g/100 mL to 5.2 g/100 mL) [44].

Another change in the composition of beverages, after the tax entered into force in Poland, was the increased content of added juices. Although the median of the juice content in NCSSBs remained at 20%, which most likely stemmed from the fact that the percentage of beverages containing ≥20% of juice was higher in 2021 by only 10.9%, a lower tax charge was still allowed. In another study carried out in Poland in 2014 on 17 fruit flavoured beverages, it was found that juices were contained in just five beverages (29%) and their corresponding amounts were only symbolic (0.1–3.0% juice). The remaining beverages used only flavourings [45]. Since the introduction of the tax, the amount of beverages containing at least 20% juices and, at the same time, no more than 5 g of sugars/100 mL which are tax-exempt, has almost tripled.

The taxation in the United Kingdom caused the proportion of red-labelled beverages to decrease from 23% to 1%, and the proportion of green-labelled beverages to increase from 6% to 27% in the group of the same beverage ranges [42]. In Poland, additional front of pack colour-coded labelling is not required. However, in accordance with the European Union law, in the case of beverages with a sugar content of ≤2.5 g/100 mL, the ‘low sugar content’ claim can be placed on the packaging [43]. Among the beverages analysed in this study, the conditions for such a nutritional claim would be fulfilled by only 4% (8/198) of beverages before the tax and only little more than 7% (14/198) after the tax.

In recent years, there has been considerable controversy around glucose-fructose syrup, which has become a frequent substitute of sucrose. The extent of its application has sparked a debate about the role of these sugars in the development of health disorders [46,47]. After the tax introduction in Poland, the frequency of sweetening beverages with glucose-fructose syrups decreased by almost three times, which should be seen as a positive change.

The author of this study is aware of its limitations. One of them is the set of beverage ranges in the five selected supermarkets in Warsaw. It is worth stressing that there are leading and large retail chains that also offer their own brands. Considering the study was carried out in the capital of Poland, the analysis included a large number of beverages available in the entire domestic market at that time. Another limitation is the fact that the sugar content was taken from the unit packages of the beverages and was not determined using laboratory methods. However, in accordance with the European Union law, manufacturers are required to share reliable data that meet the provisions on food labelling. Moreover, a study conducted previously in Poland on randomly selected soft drinks showed that the sugar content determined using high performance liquid chromatography did not differ considerably from that declared on labels [45]. Therefore, it is assumed that beverages’ nutritional values and the composition displayed on the label are consistent with the actual formulations of these products.

## 5. Conclusions

In the first year after the introduction of the sweetened beverages tax in Poland, health-enhancing changes in the composition of 62% of the beverages analysed were found. Therefore, in seeking ways to decrease the rate of tax, manufacturers are improving the nutritional value of their beverages, something which would probably have not happened to such an extent without the introduction of the tax. The reduction in the sugar content in beverages may contribute to the reduction in dietary sugar intake, while the introduction of the tax itself and the related media coverage may additionally raise the awareness of the importance of proper nutrition.

## Figures and Tables

**Figure 1 ijerph-19-14464-f001:**
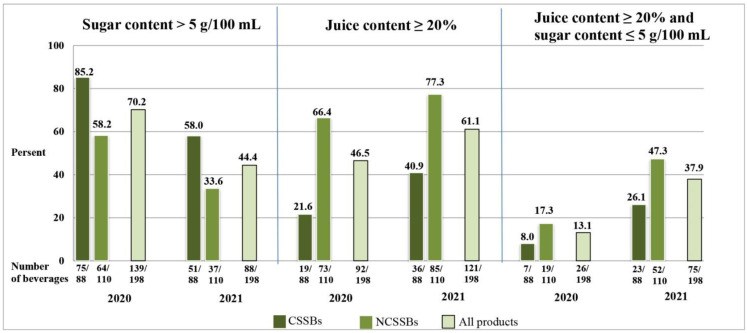
The characteristics of beverages in terms of the sweetened beverages tax criteria.

**Table 1 ijerph-19-14464-t001:** The characteristics of SSBs in both study periods.

The Parameters of Beverage Composition and Nutritional Value	2020		2021	
	CSSBs	NCSSBs	CSSBs	NCSSBs
Sugar content (g/100 mL):				
median	8.6	5.5	6.9	4.8
(min–max)	(4.2–13.0)	(1.7–13.0)	(4.2–13.0)	(1.7–11.4)
Energy value (kcal/100 mL):				
median	35.0	27.2	29.0	20.0
(min–max)	(18–52)	(11–65)	(18–52)	(1–52)
Juice content (%):				
median	1.0	20.0	3.3	20.0
(min–max)	(0–63)	(0–60)	(0–63)	(0–80)

## Data Availability

Not applicable.

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
