# Peer review of "The Impact of the Sweetened Beverages Tax on Their Reformulation in Poland—The Analysis of the Composition of Commercially Available Beverages before and after the Introduction of the Tax (2020 vs. 2021)"

_ijerph, 2022, doi:10.3390/ijerph192114464_

Round 1

Reviewer 1 Report

Dear author,

I enjoyed reading this manuscript that deals with some really practical implication of taxation of sweetened beverages and measures the impact. 

The findings are sound and clearly presented. What is missing in the intro is framework of the taxation criteria, so as it will be easier to follow the results in the context of changes of nutritional characteristics of beverages.

Author Response

Thank you very much for your kind review.

Reviewer 1.

Manuscript was corrected 

Reviewer 2.

Everything was corrected.The articles you proposed have been added

Reviewer 2 Report

Good paper, could be published. English level is acceptable.

In January 2021, Poland joined these countries where tax was imposed not only on beverages with added sugars, but also with sweeteners, regardless of the natural or synthetic origin of these substances. The range of beverages sweetened with sweeteners increased to a small extent only (by 15.2%), which can be attributed to the fact that the tax also applied to beverages with sugar substitutes. The Author should discuss why the tax applied also to beverages with sugar substitutes. To force population to drink pure water?

Some papers are missing and could be mentioned in the Introduction:

Barker AR, Mazzucca S, An R. The Impact of Sugar-Sweetened Beverage Taxes by Household Income: A Multi-City Comparison of Nielsen Purchasing Data. Nutrients. 2022 Feb 22;14(5):922. doi: 10.3390/nu14050922. PMID: 35267897; PMCID: PMC8912695.

Due to the role that sugar-sweetened beverages (SSBs) play in the obesity epidemic, SSB taxes have been enacted in the United States in the California cities of Albany, Berkeley, Oakland, and San Francisco, as well as in Boulder, Philadelphia, and Seattle. Authors pooled five years of Nielsen Consumer Panel and Retail Scanner Data (2014–18) to examine purchasing behaviors in and around these cities that have instituted SSB taxes. They included households that were either subject to the tax during the study period or were in surrounding areas within the same state. The goal was to test for the differential impact of SSB taxes by income level and type of tax. Multivariate analyses of beverage purchases found that (1) there is a dose–response relationship with the size of the SSB tax; (2) the Philadelphia tax, which is the only one that includes low-calorie beverages, is associated with greater reductions in SSB purchases and an increase in bottled water purchase; and (3) approximately 72% of the tax is passed through to consumers, but this does not vary by income level of the household. Few income-related effects were detected. Overall, the findings suggest that the Philadelphia model may be the most effective at encouraging healthy habits in beverage choice.

Andreyeva T, Marple K, Marinello S, Moore TE, Powell LM. Outcomes Following Taxation of Sugar-Sweetened Beverages: A Systematic Review and Meta-analysis. JAMA Netw Open. 2022;5(6):e2215276. doi:10.1001/jamanetworkopen.2022.15276

A total of 86 articles were eligible, with 62 studies contributing to the meta-analysis. The overall tax pass-through rate was 82% (95% CI, 66% to 98%; P < .001, I2 = 99%), suggesting tax undershifting. The demand for SSBs was highly sensitive to tax-induced price increases, with the price elasticity of demand of −1.59 (95% CI, −2.11 to −1.08; P < .001; I2 = 100%) and a mean reduction in SSB sales of 15% (95% CI, −20% to −9%; P < .001; I2 = 100%). There was no evidence of substitution to untaxed beverages, and changes in SSB consumption were not significant. The narrative synthesis found reformulation and reduced sugar content of taxed beverages for tiered taxes, cross-border shopping in most studies of local-level taxes, and no negative changes in employment.

Author Response

Thank you very much for your kind reviewes.

Everything was corrected.

The articles you proposed have been added
